# Human papillomavirus genotype and cycle threshold value from self-samples and risk of high-grade cervical lesions: A post hoc analysis of a modified stepped-wedge implementation feasibility trial

Jiayao Lei [1,2,3,4], Kate Cuschieri [5], Hasit Patel[6], Alexandra Lawrence[7], Katie Deats[1], YouScreen trial team[¶], Peter Sasieni [1☯]*, Anita W. W. Lim [4☯]

1 Centre for Cancer Screening, Prevention, and Early Diagnosis, Wolfson Institute of Population Health, Queen Mary University of London, London, United Kingdom, 2 Department of Medical Epidemiology and Biostatistics, Karolinska Institutet, Stockholm, Sweden, 3 Department of Clinical Science, Intervention and Technology, Karolinska Institutet, Stockholm, Sweden, 4 School of Cancer and Pharmaceutical Sciences, Faculty of Life Sciences and Medicine, King's College London, London, United Kingdom, 5 Scottish HPV Reference Laboratory, Dept of Laboratory Medicine, Royal Infirmary of Edinburgh & Centre for Reproductive Health, University of Edinburgh, Edinburgh, United Kingdom, 6 Health Service Laboratories LLP, Level 8 #, the Halo Building, Mabledon Place, London, United Kingdom, 7 Barts Health NHS Trust, Department of Gynaecological Oncology, Royal London Hospital, London, United Kingdom

☯ These authors contributed equally to this work.
¶ On behalf of the YouScreen Joint Steering Group (members listed in Supplementary file S1 YouScreen Trial Steering Committee). The other members of the team were Jo Gambell, Katie Deats, Dharmishta Parmar, Bernard North, and Caitlin Muller.
* p.sasieni@qmul.ac.uk

## Abstract

### Background

Human papillomavirus (HPV) testing of self-collected vaginal samples has potential to improve coverage of cervical screening programmes, but current guidelines mostly require those HPV positive on a self-sample to attend for routine screening.

### Methods and findings

A pragmatic modified stepped-wedge implementation feasibility trial was conducted at primary care practices in England. Individuals aged 25 to 64 years who were at least 6 months overdue for cervical screening could provide a self-collected sample. The primary outcomes included the monthly proportion of non-attenders screened, changes in coverage, and uptake within 90 days. Self-samples from 7,739 individuals were analysed using Roche Cobas 4800. Individuals with a positive self-sample were encouraged to attend clinical screening. In this post hoc study of the trial, we related the HPV type (HPV16, HPV18, or other high-risk type) and cycle threshold (Ct) value on the self-sample to the results of clinician-collected sample and cervical intraepithelial neoplasia grade 2 or worse (CIN2+). We wished to triage HPV–positive individuals to immediate colposcopy, clinician sampling, or

**Data Availability Statement:** Anonymous data with code book and stata code used in the main analysis are available in the Supporting Information files. The protocol of YouScreen trial is available at https://www.isrctn.com/ISRCTN12759467.

**Funding:** YouScreen is funded by the North Central London and North East London Cancer Alliance (University College Hospitals NHS Foundation Trust, https://www.uclh.nhs.uk/) through a research grant awarded to AWWL. AWWL is supported by Cancer Research UK (CRUK, https://www.cancerresearchuk.org/) grant number C8162/A16892 and C8162/A27047 awarded to PS. The Cancer Research UK & King's College London Cancer Prevention Trials Unit (CPTU) is funded by CRUK grant number C8162/A25356 awarded to PS. JL is supported by Swedish Research Council (https://www.vr.se/english.html, grant No.2021-00289 & 2023-01809) and Swedish Research Council for health, working life and welfare (https://forte.se/en/, grant No. 2023-01221). The National Institute for Health Research (https://www.nihr.ac.uk/ NIHR) covered service support costs and National Health Service commissioners funded excess treatment costs (CPMS ID: 41934). The funders had no role in study design, data collection and analysis, decision to publish, or preparation of the manuscript.

**Competing interests:** Copan Italia S.p.A provided in-kind support in the form of provision of the 552C.80 FLOQSwab for the YouScreen study. AWWL received an honorarium for a lecture from Roche (Dec 2021), and expenses to attend an expert meeting from Copan (Dec 2022). PS received an honorarium from Roche for participating in an advisory board. KC's institution has received research funding or gratis consumables to support research from the following commercial entities in the last 3 years: Abbott, Euroimmun, GeneFirst, SelfScreen, Hiantis, Seegene, Roche, Hologic, Barinthus Biotherapeutics PLC & Daye. KC has attended advisory board meetings for Hologic and Barinthus Biotherapeutics PLC, for the former, UK rail travel was paid by the company.

**Abbreviations:** ASCUS, atypical squamous cells of undetermined significance; CI, confidence interval; CSP, Cervical Screening Programme; Ct, cycle threshold; GP, General Practice; HPV, human papillomavirus; hrHPV, high-risk human papillomavirus; IQR, interquartile range; NHS, National Health Service; PCR, polymerase chain reaction.

12-month recall depending on risk. A total of 1,001 women tested positive through self-samples, and 855 women who had both an HPV–positive self-sample and a subsequent clinician-sample were included in this study. Of these, 71 (8.3%) had CIN2+. Self-sample Ct values were highly predictive of HPV in the clinician sample. Combining HPV type and Ct value allowed stratification into 3 risk groups; 44/855 (5%) were high-risk of whom 43% (19/44, 95% confidence interval [29.7%, 57.8%]) had CIN2+. The majority (52.9%, 452/855) were low-risk, of whom 4% (18/452, 95% CI [2.5%, 6.2%]) had CIN2+. The main limitation of our study was the colposcopy assessment was restricted to individuals who had abnormal cytology after positive results of both self-sample and clinician-collected sample.

## Conclusions

HPV type and Ct value on HPV–positive self-samples may be used for triage. The difference in the risk of CIN2+ in these groups appears sufficient to justify differential clinical management. A prospective study employing such triage to evaluate laboratory workflow, acceptability, and follow-up procedure and to optimise clinical performance seems warranted.

## Trial Registration

ISRCTN12759467.

## Author summary

### Why was this study done?

- Human papillomavirus (HPV) testing of self-collected vaginal samples has potential to improve coverage of cervical screening programmes, but current guidelines mostly require those HPV positive on a self-sample to attend for routine screening.
- The association between HPV cycle threshold (Ct) values (as a proxy for viral load), HPV genotypes, and the risk of high-grade cervical precancerous lesions and cancers has been well established. However, most of the existing studies are based on practitioner-collected cervical samples, with limited research focusing on self-collected samples.

### What did the researcher do and find?

- We propose a stratification approach, dividing HPV–positive individuals into 3 distinct risk groups based on HPV Ct values and genotypes (HPV16, HPV18, or other high-risk type) from HPV self-samples.
- We classified 5% (44/855) of the individuals (HPV-16 Ct <30) who tested HPV-positive on their self-sample as high-risk, with 43.2% (19/44) of them having CIN2+, which is comparable to those referred in England with HPV positive and abnormal cytology results.

- The low-risk group (HPV non-16/18 Ct $\geq$ 30), comprising half of those with HPV on their self-sample have a risk of CIN2+ of 4.0% (18/452) which is lower than in those who are HPV positive but cytology normal on a routine clinician screen.

## What do these findings mean?

- The proposed classifier allows those at greatest risk (high-risk group) to be sent directly to colposcopy and detects 3 quarters of CIN2+ without delay while allowing half of individuals (low-risk group) to be managed by repeat self-sampling. The intermediate risk group could be referred to their general practises for clinical sampling.

- Findings support the potential use of this risk classifier in the management of HPV–positive self-samples within organised cervical screening programmes.

- Limitations of the study include that the colposcopy assessment was restricted to individuals who had abnormal cytology after positive results of both self-samples and clinician-collected samples. Additionally, different HPV assays were used for primary screening and follow-up tests.

## Introduction

Cervical cancer remains a significant global health concern, with screening being one of the key pillars in the global strategy of eliminating cervical cancer [1]. Evidence from randomised clinical trials and real-world cervical screening programmes have demonstrated that human papillomavirus (HPV) testing offers more protection than cytology [2–4]. HPV-based cervical screening has become standard worldwide [5,6]. HPV testing on self-collected cervicovaginal material (HPV self-sampling) has shown promise in increasing screening participation, especially among under-screened populations [7–10]. Self-samples have also demonstrated either a comparable [7] or slightly lower [11] clinical sensitivity and specificity for detecting high-grade cervical lesions, providing a valuable tool for early detection and prevention of cervical cancer. In a routine programme, where complete disease verification for all screened individuals is not feasible, the predictive value of a positive-screen for high-grade cervical lesions is important in informing various clinical management pathways.

HPV assays based on polymerase chain reaction (PCR) chemistry and other target amplification assays often provide a semiquantitative output known as the cycle threshold (Ct). Ct is inversely related to the amount of target in a reaction and can offer a rough proxy of viral load in those assays which report it. The relationship between HPV viral load and risk of cervical disease is slightly ambivalent in the literature but there are reports that type-specific viral load may be informative for risk stratification, particularly for certain types including HPV 16 [11–16]. To the best of our knowledge, risk stratification models that incorporate Ct values and HPV genotypes from self-samples have not been developed, particularly not within the context of United Kingdom (UK) screening programmes.

HPV self-sampling has been integrated in routine screening programmes of several countries to reach under-screened individuals or as an opt-in alternative for individuals who prefer self-sampling to clinician-collected samples [17–20]. In most such programmes, individuals

who have HPV detected on their self-sample must attend for clinician sampling to support triage testing (which often includes cytological/triage assessment); this can lead to a loss to follow-up and is associated with patient and clinic time. Alternative triage approaches that can use the original self-sample would mitigate some of these issues.

The aim of our study is to examine the association between type-specific viral load (i.e., HPV genotype and Ct value) on HPV self-samples and risk of disease to gain insight into their suitability as triage strategies. Specifically, we assessed association of type and Ct value and the risk of being HPV-negative on the clinical sample or of high-grade cervical lesions detected at colposcopy. This analysis was based on data accumulated from the YouScreen trial of self-sampling in London, UK [10]. We assess the possibility of including both load- and type-based indicators for clinical management in individuals who test positive for high-risk HPV (hrHPV, i.e., HPV types that can cause cervical cancer) on self-collected vaginal swabs. We followed Castle and Katki [21] who advocate "equal management for equal risks" as a framework for developing screening guidelines and practice, and envisage 3 management options for HPV-positive individuals based on the self-sample alone: direct referral to colposcopy (high risk), referral for clinical sampling in primary care (medium risk), repeat self-sampling at 12 months (low risk).

## Materials and methods

### Study setting and population

In England, the National Health Service (NHS) offers cervical screening to individuals aged 25 to 64 years with a cervix, primarily conducted in General Practice (GP) clinics using HPV testing as a primary screen with cytology triage. The program is managed through a central database (NHAIS), which administers invitations on a 3- (25 to 49 years) or 5-year (50 to 64 years) basis, depending on the age group. Currently, all routine cervical screening in England is based on clinician collection.

The YouScreen trial [10] was a pragmatic, modified stepped-wedge implementation feasibility clinical trial nested within the English Cervical Screening Programme (CSP) involving 133 participating and 62 non-participating GP clinics. It was conducted in GP clinics within the 5 London boroughs with some of the lowest cervical screening coverage rates in the country. It aimed to enhance screening attendance by offering HPV self-sampling kits to individuals who had not engaged with the routine offer of cervical screening. The offer of self-sampling kits was made either through direct mail or opportunistically by the physician or nurse when the patient consulted with their GP practice between January 13, 2021 and November 30, 2021. The target group comprised individuals aged 25 to 64 years who were at least 6 months overdue for their routine cervical screening. All HPV testing of self-samples was done using the Roche cobas 4800 HPV assay (Roche Diagnostics GmBH). Clinician-taken follow-up liquid-based samples were analysed using APTIMA (Hologic, Manchester). Individuals who tested HPV–positive on their clinician sample had reflex cytology. Those who tested HPV–positive and had any abnormality on cytology were referred to colposcopy. Details were described previously [10] and the trial is registered with ISRCTN:12759467.

Ct values are not readily available from cobas 4800 and there was no plan to analyse them in the YouScreen protocol. When the trial was complete, we managed to extract Ct values with specific support from the manufacturer and planned the post hoc study reported here. Our study population included individuals from the YouScreen trial with an HPV–positive self-sample and a follow-up clinician-collected sample with a valid HPV result. We collected and analysed the HPV testing outcomes from both self-samples and clinician-collected samples, along with their cytological and histopathological evaluations where a biopsy had been routinely indicated.

## Laboratory testing

The self-sampling kits offered were 552C.80 FLOQswab flocked swab (Copan Italia, Brescia, Italy). Samples were transported dry at room temperature. The dry swabs were suspended in 5 ml of PreservCyt media (Hologic, USA) and vortexed for 10 s in the original sample tube. These tubes were then stored at 5˚C until testing. An aliquot was transferred to another tube and loaded onto the Roche Cobas 4800 system for analysis. This resuspension volume is consistent with that used for self-sampling in national programmes including in the Netherlands and Australia. The Roche cobas 4800 HPV assay individually identifies (types) HPV16 and HPV18 while detecting 12 other high-risk types (31, 33, 35, 39, 45, 51, 52, 56, 58, 59, 66, and 68) as a pool. Ct values were reported for HPV16, HPV18, and non-16/18 hrHPV. The Cobas assay simultaneously tests for human beta-globin as an internal control for specimen cellularity. The manufacturers have defined Ct value-cutoffs for all the HPV channels to determine positive results (the Ct value must be below the cut-off for the test to be called positive given that lower Ct values tend to indicate higher viral load), which are 40.5 for HPV16 and 40 for HPV18 and the pooled 12-HPV channels. Ct values for positive HPV results and for beta-globin were obtained from the cobas machine. Women for whom Ct values were not available were not included in this analysis.

Individuals who received a positive HPV result from their self-samples were recommended to undergo a clinician-collected follow-up test (primary HPV screening test) which was managed according to the routine cervical screening guidelines [22]. Individuals who tested negative for HPV on their clinician sample were considered to have screened negative and were returned to routine screening. Fig 1 contains details of sample flow.

HPV–positive clinician-collected samples were used for reflex cytology. If an abnormal cytology (atypical squamous cells of undetermined significance (ASCUS) or worse) was present, the patient was referred to colposcopy. Colposcopy and, when a biopsy was taken, histology outcomes were retrieved from the Cyres database which aggregates data from all NHS colposcopy clinics across London. Since all histopathological diagnoses were obtained through the routine programme, there was no central pathological review. Individuals with normal cytology would have been recalled for further HPV testing at 12 months. Results of that "early recall" are not included in this study. Data linkage was accomplished using an encrypted version of each woman's unique personal identifier through OpenPseudonymiser (https://www.openpseudonymiser.org/Default.aspx). The outcome from the histopathological diagnoses were classified as cervical intraepithelial neoplasia grade 2 or worse (CIN2+) and cervical intraepithelial neoplasia grade 3 or worse (CIN3+) in our study.

## Statistical analyses

When considering all hrHPV types combined, the overall HPV Ct value was defined as the lowest Ct value of HPV16, HPV18, or other hrHPV types. Among individuals who tested positive for hrHPV and had a follow-up clinician-collected sample, we described the distribution of HPV Ct values in self-collected samples and beta-globin Ct values by HPV positivity and the follow-up cytological results in clinician-collected samples through box and whisker plots.

We determined the proportions of (i) individuals with HPV–positive clinician-collected samples; (ii) individuals referred to colposcopy after the HPV–positive clinician-collected samples; and those found to have (iii) CIN2+ and (iv) CIN3+. We anticipated that the probability of (i) being HPV–positive on a clinician-collected sample; (ii) being referred to colposcopy after clinical sampling; and (iii) of finding CIN2+ or CIN3+ on colposcopy would all be related to which of the 3 HPV channels (16, 18, other high-risk type) was positive and inversely related to the Ct value of that channel and it would be largely unrelated to the Ct value of the beta-

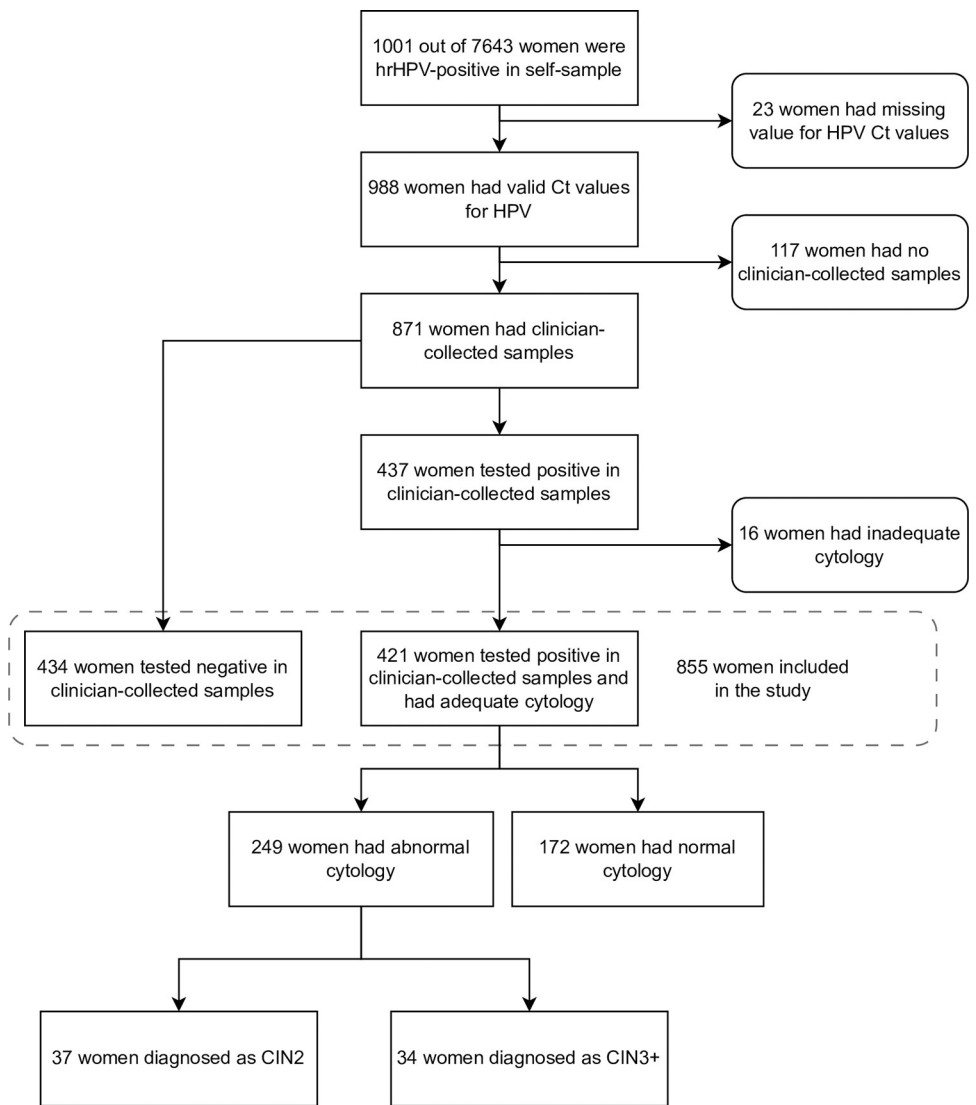

**Fig 1. Study participants included in the study.** 769 out of 855 (90.0%) women had clinician-collected samples within 3 months after the positive HPV self-samples. hrHPV, high-risk human papillomavirus; HPV, human papillomavirus; Ct, cycle threshold; CIN2, cervical intraepithelial neoplasia grade 2; CIN3+, cervical intraepithelial neoplasia grade 3 or worse.

globin channel. Our model assumes that the outcomes follow binomial distributions with specified probabilities. While there is no explicit formal relationship between the different outcomes, they are nested (i.e., HPV positivity in clinician-collected samples > abnormal cytology > CIN2+ > CIN3+, where "A > B" indicates that all individuals with "B" also have "A"). The primary focus of the modelling is the CIN2+ outcome.

Given age affects the risk of CIN2+ and CIN3+ in screening and it is common in the literature to encounter references to the "Ct value" without specifying the associated HPV type, these proportions were stratified by age (categorized as <30 years, 30 to 49 years and 50+ years to reflect age-dependent levels of HPV prevalence and screening intervals) and the overall HPV Ct value categories (<30 30 to 35, and 35+). The cut-offs for Ct values were chosen to produce roughly equal size of groups and because they had been used in a published study

[11]. We also calculated these proportions by a combination of (i) HPV types and Ct values, and (ii) age (categorised as <50 years and 50+ years) and overall HPV Ct values. The classification of HPV types was done hierarchically, prioritising HPV 16 followed by HPV 18, and then other hrHPV types, because the risk of CIN2+ is greatest for HPV16 positive and next greatest for HPV18 positive.

Finally, we used fractional polynomial regression [23–25] in which multiple models are fit using different pairs of power transformation and the best fitting pair is selected, to examine the association between risk of CIN2+ and overall HPV Ct values. We predicted the risk (probability and 95% confidence interval (CI)) of CIN2+ as a function of the Ct values. We repeated the analysis separately for HPV16 and non-HPV16 Ct values to assess differences in the risk of CIN2+.

In a sensitivity analysis, we tested the robustness of our model by applying various smoothing methods for HPV Ct values and the probabilities of CIN2+. Specifically, we employed the following approaches in Stata: (i) running mean smoothing using the command *running*; (ii) natural cubic splines using the Stata command *makespline* in a logistic regression model with 4 knots; (iii) natural cubic splines using the command *spline*. We then compared and plotted the predicted probabilities of CIN2+ for each smoothing method based on the overall HPV Ct values, Ct values of HPV 16, and Ct values of non-HPV16. Additionally, we compared the 95% CIs of the predicted probabilities obtained from fractional polynomial regression and logistic regression models. We used Stata (version 17.0) for data management and statistical analyses.

## Target risk classification

The management of all HPV–positive individuals in YouScreen was referral to primary care for clinician sampling. We wanted to identify a small high-risk group who might be referred directly to colposcopy, and, if possible, a large low-risk group in whom repeat self-sampling at 12 months might be sufficient. Referral pathways in cervical screening are ideally risk-based [26]. We based the target specifications of this risk classifier on the risk of CIN2+ with reference to the current screening programme in England. For immediate referral to colposcopy (high-risk), we wanted the risk to be similar or greater than that of those currently referred to colposcopy. For repeat testing at 12 months rather than immediate clinical sampling, we wanted the risk to be no more than the risk of CIN2+ in those currently offered early recall. In the HPV primary screening pilot [3], among those referred "immediately" with HPV–positive and abnormal cytology, about 41% (3,060/7,542) had CIN2+. Whereas of those referred at 24 months (after persistent HPV–positive but with normal cytology at baseline and 12 months) 14% (261/1,912) had CIN2+. Additionally, of those who at baseline were HPV–positive and cytology normal who had a repeat screen at 12 months as recommended, 8.3% (1059/12,830) were found to have CIN2+ either at 12 or 24 months. Of those screened by HPV testing (on the prevalent round), 2.3% (4,156/183,970) were found to have CIN2+ in the first round. Thus, we required the prevalence of CIN2+ in the high-risk group to be at least 14% and ideally around 40% in accordance with proportion of CIN2+ detected based on current referral pathways. Similarly, the low-risk group should have at most 8% and ideally about 2.3% with CIN2+. We computed 95% Wilson Score CIs for the proportion of CIN2+ and CIN3+ in each of the risk group.

## Ethics

Ethical approval for the YouScreen clinical trial was granted by the South Birmingham Research Ethics Committee (20/WM/0120), IRAS ID 264776. Additional approvals were received from the Confidentiality Advisory Group (20/CAG/0086) and the Cervical Screening

Programme's Research Advisory Committee (CSP-RAC-032). All participants in this study provided informed consent implicitly when they returned a sample for HPV testing.

## Results

### Study population

Of the 7,643 individuals who had self-samples, 1,001 had a positive result for hrHPV (13.1%) (Fig 1). Among those we were able to obtain HPV Ct values for 988 individuals of whom 871 (88.2%) had a follow-up clinician-collected sample. The number of individuals who tested HPV–positive on their clinician-collected samples after the positive self-samples was 437/871 (50.2%). Among those HPV–positive on clinical-collected samples, 421/437 (96%) individuals had adequate cytology including 249/421 (59.1%) with abnormal results. Subsequently, 37 individuals were diagnosed with CIN2 and 34 with CIN3+: corresponding to 71 individuals with CIN2+ or 28.5% (71/249) of those referred to immediate colposcopy. The majority of included individuals were: aged 25 to 39 years (69.2%, 592/855); late for screening between 6 months and 24 months (51.4%, 439/854); of white ethnicity (48.6%, 313/644); and from the 2 most deprived quintiles (60.0%, 512/855) (S1 Table).

### HPV and beta-globin Ct values

Beta-globin Ct values (on self-samples) were very similar between those who were HPV–negative, HPV–positive with normal cytology, and HPV–positive and abnormal cytology on their clinical samples (Fig 2). The medians were 26.9 (interquartile range (IQR): 25.8 to 28.4) for HPV–negative, 26.5 (IQR: 25.4 to 28.3) for HPV–positive with normal cytology, and 26.6 (IQR: 25.6 to 27.9) for HPV–positive with abnormal cytology. Notably, there were many outliers (Ct > 32.5) for beta-globin Ct values, indicating samples with low levels of human DNA, in all groups.

For individuals who tested HPV–negative on their clinician-collected sample, the median HPV Ct value on their self-sample was 36.7 (IQR: 32.8 to 38.7), while the median HPV Ct values were 30.9 (IQR: 26.7 to 35.4) for individuals tested HPV–positive with normal cytology and 28.6 (IQR: 25.0 to 33.3) for individuals tested HPV–positive with abnormal cytology.

### HPV types, HPV Ct values, and risk of high-grade cervical lesions

The proportion of CIN2+ and CIN3+ in individuals with HPV–positive self-sample by age, HPV type, and HPV Ct value are presented in Table 1. Overall, 71 (8.3%) of 855 individuals had CIN2+ and 34/855 (4.0%) had CIN3+. Stratifying by age, the proportion with CIN2+ and CIN3+ were much lower among individuals aged above 50 years (3.4% [4/116] and 1.7% [2/116]), and the proportions were comparable for women aged below 30 years (9.0% [24/266] and 4.1% [11/266]) and aged 30 to 49 years old (9.1% [43/473] and 4.4% [21/473]).

Comparing by HPV Ct values, we observed a clear trend of decreased HPV positivity on clinician-collected samples with increasing Ct levels: 80.1% (218/272, 95% CI [75.0%, 84.4%]), 52.5% (114/217, 95% CI [45.9%, 59.1%]), and 24.3% (89/366, 95% CI [20.2%, 29.0%]) for individuals with HPV Ct of below 30, 30 to 35, and above 35, respectively. The corresponding percentages with CIN2+ were 14.3% (39/272), 7.4% (16/217), and 4.4% (16/366), respectively. The risk of CIN2+ plotted as a smooth function of the HPV Ct value is shown in Fig 3. Although the proportion with CIN2+ is below 10% for Ct values below 20, there are very few observations with such low Ct values and the 95% CIs includes both low and high probabilities of CIN2+. It increases to nearly 15% for Ct values in the mid-20s and then gradually falls dipping below 5% at about 35 and below 2.5% by 40. The pattern was consistent when stratifying by

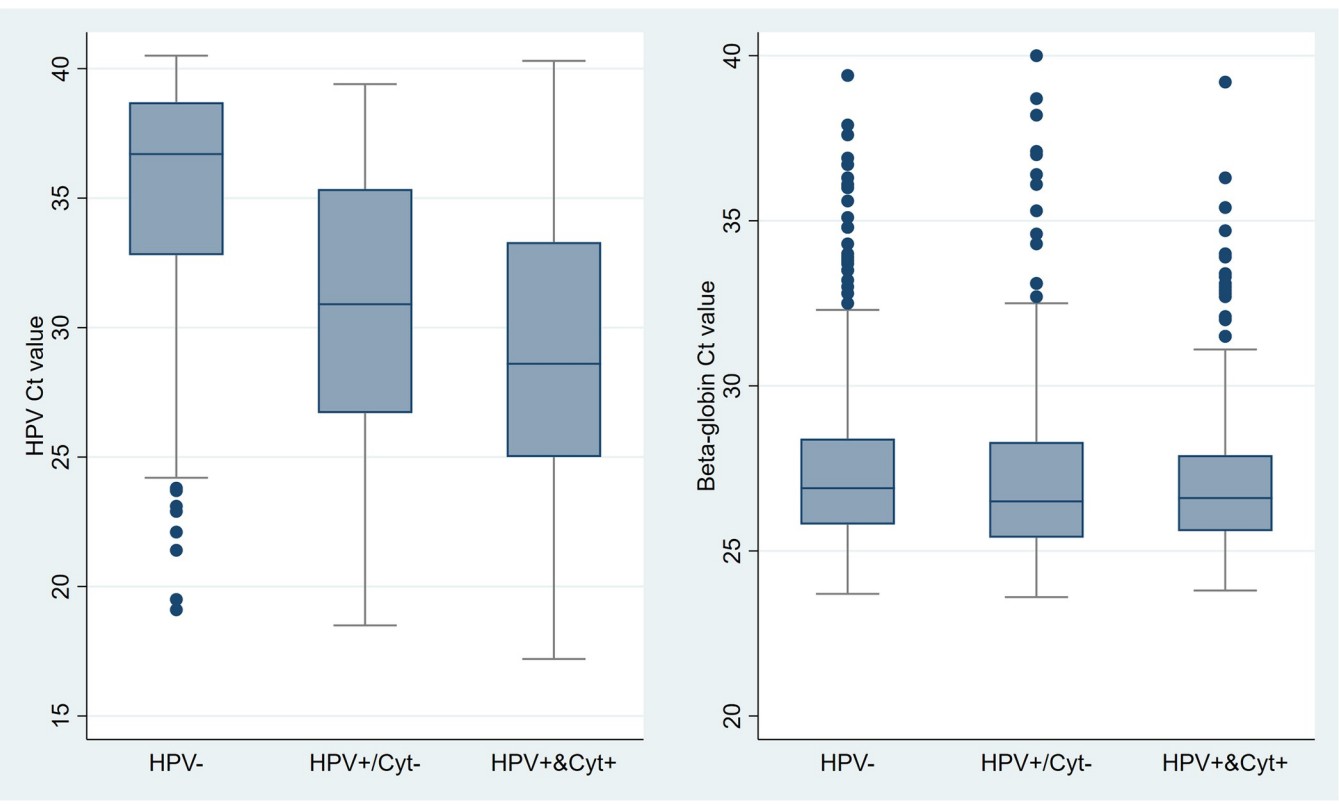

**Fig 2. Box and whisker plot of HPV Ct values and beta-globin Ct values in self-samples by HPV and cytological results in clinician-collected samples.** Beta-globin serves as an internal control for specimen cellularity. HPV, human papillomavirus; Ct, cycle threshold. Cyt+ refers to abnormal cytology including atypical squamous cells of undetermined significance or worse. Cyt- refers to normal cytology.

age and the magnitude of risk for each of level of Ct values was higher for individuals below age 50 years.

Looking at both HPV type and Ct values, of individuals who were HPV 16–positive with an HPV Ct value below 30, 43.2% (19/44) had CIN2+ and 31.8% (14/44) had CIN3+. The proportions with disease were lower among those who had an HPV-16 Ct value between 30 and 35 but remained high (19.0% had CIN2+ and 7.1% had CIN3+). For individuals HPV16–positive and with a Ct value above 35, 9.2% and 4.6% had CIN2+ and CIN3+, respectively. Among the relatively small numbers of individuals who tested positive for HPV18, the proportions with CIN2+ were 12.5%, 0%, and 12.0% for those with Ct values below 30, between 30 and 35, and above 35, respectively. For individuals with other hrHPV–positive and Ct value above 30, the proportion with CIN2+ was 7.8%, 5.0%, and 3.4%, while the proportion with CIN3+ was 1.5%, 1.9%, and 2.1%, respectively.

We also modelled the risk of CIN2+ against HPV Ct values separately for HPV-16 and non-16 HPV. The risk of CIN2+ for individuals who were HPV-16–positive was much more pronounced and fell monotonically with increase Ct value. It was over 40% for those who had a Ct value of <25 and was still over 10% for Ct of 35. By contrast, the risk of CIN2+ for individuals positive for non-16 HPV peaked at around 10% across all range of Ct values and was lower both for very low and very high Ct values (Fig 4). In the sensitivity analyses applying varying degrees of smoothing, the predicted probabilities of CIN2+ were comparable across different smoothing methods, and generally showed within the 95% confidence bands from the fractional polynomial regression (S1 Fig). The 95% confidence bands from the logistic

**Table 1. Proportions with HPV positivity in clinician-collected samples, referral to colposcopy, CIN2+ and CIN3+ by age and Ct values.**

| | N = 855[a] | hrHPV+ in CC samples | | Referred to colposcopy[b] | | CIN2+ | | CIN3+ | |
|---|---|---|---|---|---|---|---|---|---|
| | | N | % | N | % | N | % | N | % |
| **Age** | | | | | | | | | |
| Age <30 years | 266 | 148 | 55.6% | 89 | 33.5% | 24 | 9.0% | 11 | 4.1% |
| Age 30–49 years | 473 | 235 | 49.7% | 139 | 29.4% | 43 | 9.1% | 21 | 4.4% |
| Age 50+ years | 116 | 38 | 32.8% | 21 | 18.1% | 4 | 3.4% | 2 | 1.7% |
| **HPV Ct value** | | | | | | | | | |
| <30 | 272 | 218 | 80.1% | 147 | 54.0% | 39 | 14.3% | 18 | 6.6% |
| 30–35 | 217 | 114 | 52.5% | 59 | 27.2% | 16 | 7.4% | 6 | 2.8% |
| 35+ | 366 | 89 | 24.3% | 43 | 11.7% | 16 | 4.4% | 10 | 2.7% |
| **HPV 16, 18, and other types** | | | | | | | | | |
| **HPV 16 Ct value** | | | | | | | | | |
| <30 | 44 | 41 | 93.2% | 36 | 81.8% | 19 | 43.2% | 14 | 31.8% |
| 30–35 | 42 | 24 | 57.1% | 14 | 33.3% | 8 | 19.0% | 3 | 7.1% |
| 35+ | 65 | 25 | 38.5% | 14 | 21.5% | 6 | 9.2% | 3 | 4.6% |
| **HPV 18 Ct value** | | | | | | | | | |
| <30 | 8 | 7 | 87.5% | 4 | 50.0% | 1 | 12.5% | 1 | 12.5% |
| 30–35 | 14 | 7 | 50.0% | 6 | 42.9% | 0 | 0.0% | 0 | 0.0% |
| 35+ | 25 | 7 | 28.0% | 5 | 20.0% | 3 | 12.0% | 1 | 4.0% |
| **Non-16/18 HPV Ct value** | | | | | | | | | |
| <30 | 205 | 156 | 76.1% | 97 | 47.3% | 16 | 7.8% | 3 | 1.5% |
| 30–35 | 160 | 86 | 53.8% | 41 | 25.6% | 8 | 5.0% | 3 | 1.9% |
| 35+ | 292 | 68 | 23.3% | 32 | 11.0% | 10 | 3.4% | 6 | 2.1% |
| **Age and HPV Ct value** | | | | | | | | | |
| **Age <50 years, HPV Ct value** | | | | | | | | | |
| <30 | 254 | 205 | 80.7% | 137 | 53.9% | 38 | 15.0% | 18 | 7.1% |
| 30–35 | 191 | 106 | 55.5% | 55 | 28.8% | 15 | 7.9% | 5 | 2.6% |
| 35+ | 294 | 72 | 24.5% | 36 | 12.2% | 14 | 4.8% | 9 | 3.1% |
| **Age 50+ years, HPV Ct value** | | | | | | | | | |
| <30 | 18 | 13 | 72.2% | 10 | 55.6% | 1 | 5.6% | 0 | 0.0% |
| 30–35 | 26 | 8 | 30.8% | 4 | 15.4% | 1 | 3.8% | 1 | 3.8% |
| 35+ | 72 | 17 | 23.6% | 7 | 9.7% | 2 | 2.8% | 1 | 1.4% |
| **Total** | 855 | 421 | 49.2% | 249 | 29.1% | 71 | 8.3% | 34 | 4.0% |

[a]Only included women who had adequate cytology.

[b]Only women who had abnormal cytology were referred to colposcopy.

HPV, human papillomavirus; hrHPV, high-risk human papillomavirus; Ct, cycle threshold; CC, clinician-collected; CIN2+, cervical intraepithelial neoplasia grade 2 or worse; CIN3+, cervical intraepithelial neoplasia grade 3 or worse.

regressions using fractional polynomials and restricted cubic splines also show substantial overlap.

## Risk groups

Based on the above, we divided HPV–positive self-samples into 3 levels of risk (Tables 1 and 2).

Under{High risk}: HPV-16 Ct <30. This group included 44 individuals (5% [44/855] of all the HPV–positive), of whom 19 (43.2%, 95% CI [29.7%, 57.8%]) had CIN2+ and 14 (31.8%, 95% CI [20.0%, 46.6%]) had CIN3+. Seeing that the percentage for CIN2+ exceeds the upper target

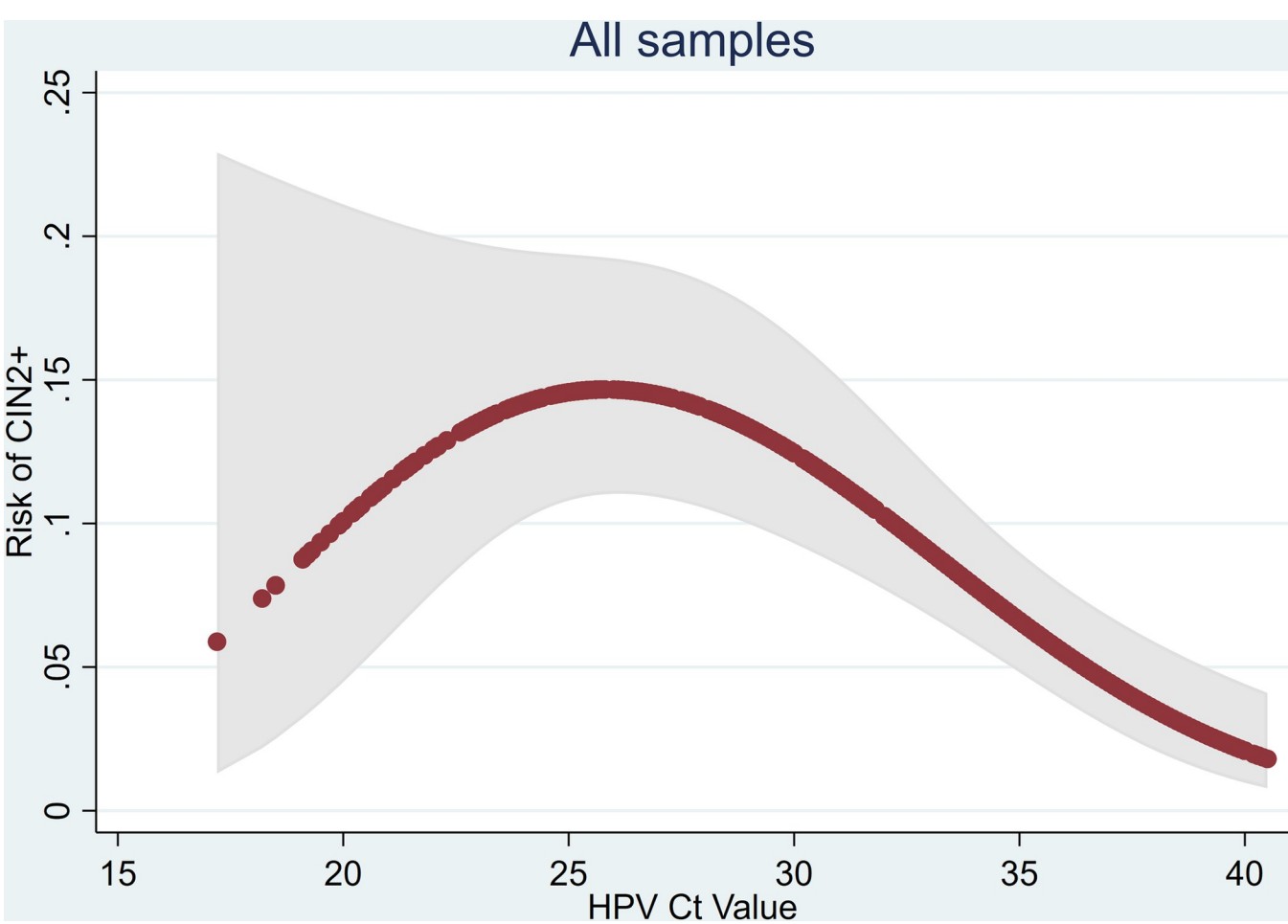

**Fig 3. Risk of CIN2+ and HPV Ct value in all samples.** Risk estimated by fitting a fractional polynomial in a logistic regression model (red dots) together with a 95% confidence band (grey shading). HPV, human papillomavirus; Ct, cycle threshold; CIN2+, cervical intraepithelial neoplasia grade 2 or worse.

of 40% and greatly exceeds the lower target of 14%, we recommend that they are referred to colposcopy immediately. We note that despite referring first to the GP for clinician sampling, 81.8% (36/44) were referred to immediate colposcopy after clinician sampling. A further 11% (5/44) were HPV–positive on the clinician sample and may be referred to colposcopy at 12 or 24 months depending on the results of their early recall.

Intermediate risk: HPV-16 Ct ≥30 or HPV-18 (any Ct value) or HPV non16/18 Ct <30. This group includes 359 individuals (42% of 855 HPV–positive individuals), of whom 34/359 (9.5%, 95% CI [6.9%, 12.9%]) had CIN2+ and 11/359 (3.1%, 95% CI [1.7%, 5.4%]) had CIN3+. The proportion with CIN2+ is too low to recommend immediate referral to colposcopy but too high to wait for 12 months. We recommend that they are referred to their GP for clinical sampling where (in this study) 39% (140/359) were subsequently referred to immediate colposcopy.

Low risk: HPV non-16/18 Ct ≥30. This group includes 452 individuals (53% of 855 HPV–positive individuals), of whom 18/452 (4.0%, 95% CI [2.5%, 6.2%]) had CIN2+ and 9/452 (2.0%, 95% CI [1.1%, 3.7%]) had CIN3+. Since even the upper limit of the 95% CI for CIN2+ is less than 8%, we suggest that they were recommended to have a repeat self-sample in 12 months. Note that after clinical sampling, only 16% (73/452) were subsequently referred to colposcopy in this study.

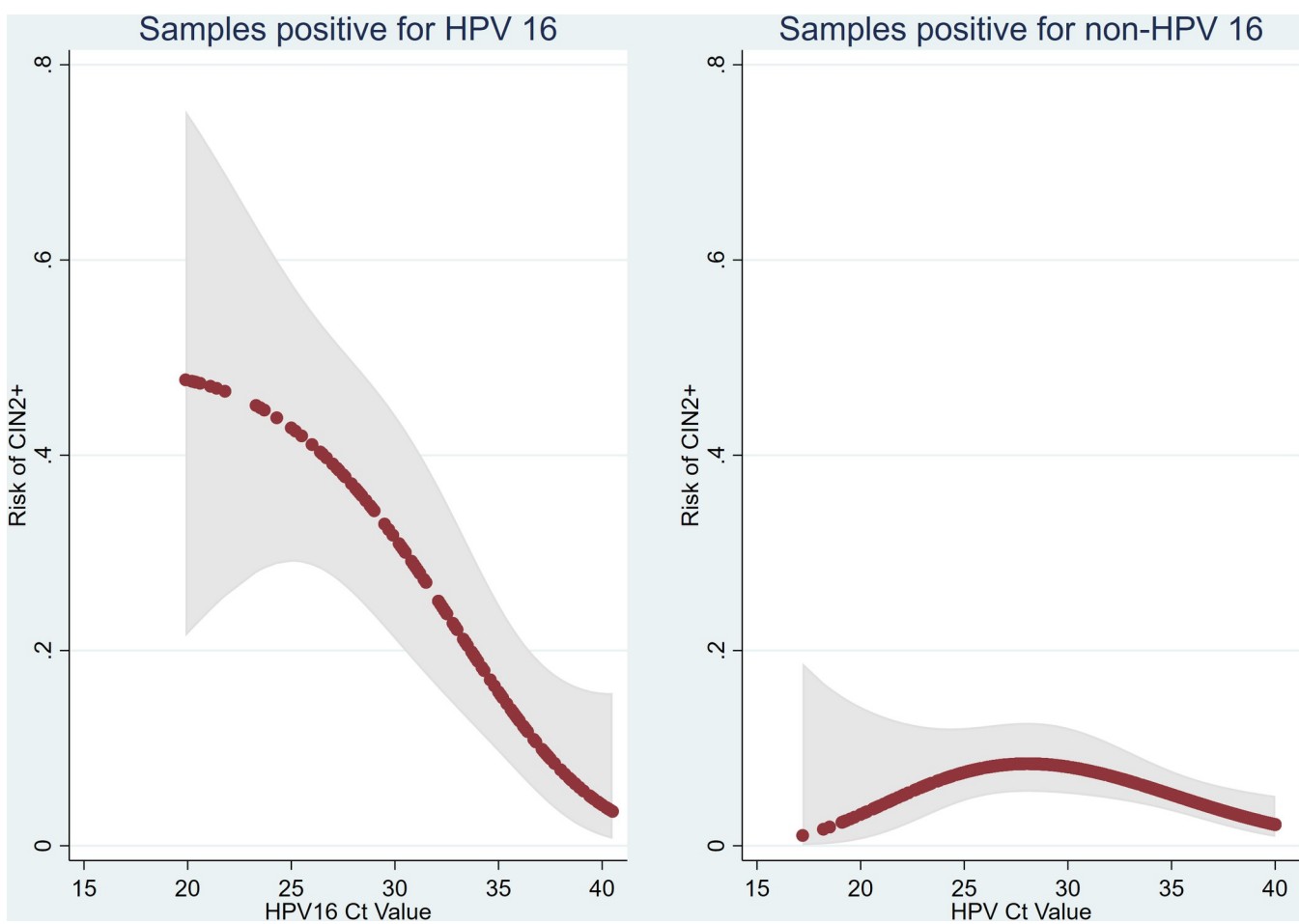

**Fig 4. Risk of CIN2+ and HPV16 Ct value in HPV 16–positive samples and non-HPV 16–positive samples.** Risk estimated by fitting a fractional polynomial in a logistic regression model (red dots) together with a 95% confidence band (grey shading). HPV, human papillomavirus; Ct, cycle threshold; CIN2+, cervical intraepithelial neoplasia grade 2 or worse.

**Table 2. Proportions with HPV positivity in clinician-collected samples, referral to colposcopy, CIN2+ and CIN3+, in each level of our risk groups.**

| Risk groups[a] | | hrHPV+ in CC samples | | Referred to colposcopy[c] | | CIN2+ | | CIN3+ | |
|---|---|---|---|---|---|---|---|---|---|
| | N = 855[b] | N | % | N | % | N | % (95% CI) | N | % (95% CI) |
| High risk | 44 | 41 | 93.2% | 36 | 81.8% | 19 | 43.2% (29.7%, 57.8%) | 14 | 31.8% (20.0%, 46.6%) |
| Intermediate risk | 359 | 226 | 63.0% | 140 | 39.0% | 34 | 9.5% (6.9%, 12.9%) | 11 | 3.1% (1.7%, 5.4%) |
| Low risk | 452 | 154 | 34.1% | 73 | 16.2% | 18 | 4.0% (2.5%, 6.2%) | 9 | 2.0% (1.1%, 3.7%) |
| **Total** | **855** | **421** | **49.2%** | **249** | **29.1%** | **71** | **8.3% (6.6%, 10.3%)** | **34** | **4.0% (2.9%, 5.5%)** |

[a] High risk group includes individuals had HPV-16 Ct <30. Intermediate risk group includes individuals had HPV-16 Ct ≥30 or HPV-18 (any Ct value) or HPV non16/18 Ct <30. Low risk group includes individuals had HPV non-16/18 Ct ≥30.

[b] Only included women who had adequate cytology.

[c] Only women who had abnormal cytology were referred to colposcopy.

HPV, human papillomavirus; CC, clinician-collected; CIN2+, cervical intraepithelial neoplasia grade 2 or worse; CIN3+, cervical intraepithelial neoplasia grade 3 or worse; CI, confidence interval.

Individuals in the intermediate risk group have slightly higher risk of having CIN2+ but a slightly lower risk of having CIN3+ compared to all HPV–positive individuals. The risk for CIN2+ is 9.5% (95% CI [6.9%, 12.9%]) versus 8.3% (95% CI [6.6%, 10.3%]) for all HPV–positive individuals, while the risk for CIN3+ is 3.1% (95% CI [1.7%, 5.4%]) versus 4.0% (95% CI [2.9%, 5.5%]). By contrast, the individuals in the low-risk group had about half the risk (4.0%, 95% CI [2.5%, 6.2%] versus 8.3%, 95% CI [6.6%, 10.3%] and 2.0%, 95% CI [1.1%, 3.7%] versus 4.0%, 95% CI [2.9%, 5.5%], respectively). Using this classifier allows those at greatest risk to be sent directly to colposcopy and detects 3 quarters of CIN2+ immediately while allowing half of individuals to be managed by repeat self-sampling.

## Discussion

In this study, HPV Ct values from self-samples were associated with the risk of CIN2+ and CIN3+, supporting the potential use of HPV Ct value and HPV genotypes to guide the clinical management of HPV–positive results in self-samples. Individuals with lower HPV Ct values on their self-samples (higher viral load) were at higher risk of CIN2+ and CIN3+. The risk of CIN2+ and CIN3+ also depended on the HPV genotype. The risk of CIN2+ was greater in individuals positive for HPV16 than in individuals positive for other types.

As discussed previously [27], the proportion (50%) of clinician-collected samples that were HPV–negative in women with an HPV–positive self-sample was unexpectedly low. However, it was very similar to that reported (55.2%) in a large study from Sweden [28]. Of note, although Ct values for beta-globin over 32.5 are outliers on the box and whiskers plots, they were all within the manufacturer's adequate range. Likely explanations for the lack of concordance between the self- and clinician-samples include: vaginal infections that were not present in the cervix [7], an HPV deposition rather than a true infection [29], difference in false positives between Roche cobas 4800 and Aptima as well as viral clearance between collection of the first and second samples [30].

When examining the combination of HPV genotype and Ct level, we observed that the intermediate risk group could be further divided in 2 subgroups. A small high-intermediate group would include individuals who were HPV16–positive with Ct value of 30 to 35 or HPV18–positive with Ct value below 30 (18% [9/50] of whom had CIN2+). The remaining low-intermediate risk group HPV16–positve Ct ≥35, HPV18–positive Ct ≥30, or had other hrHPV Ct <30 (8% [25/309] of whom has CIN2+). However, we think that dividing HPV–positive individuals into 4 risk groups may introduce noise by being overly reliant on the data in this study. We therefore prefer to divide into 3 risk groups as described in the Results section. Additionally, we note that the risk of CIN2+ among individuals aged over 50 years in the intermediate risk group was only 2.4% (1/42) (S2 Table). Thus, one might reclassify individuals aged 50+ years positive for HPV non-16/18 as low risk regardless of their Ct value.

The pattern observed in Fig 3 is not as expected. Rather than observing a monotone decrease in risk with increasing Ct values, we see a bell-shaped curve with a mode at a Ct value of about 27. There are 3 potential explanations for this. Firstly, this graph includes Ct values from all 3 HPV channels. When restricted to HPV16 (Fig 4), the curve is monotone. The majority of Ct values below 20 are from non-HPV16 types that have a lower risk of CIN2+. Nevertheless, the plot for non-HPV16 Ct values is also bell shaped. Interestingly, Zhang and colleagues [14] also observed a strong monotone association for HPV16 and a fairly flat relationship for HPV18 and other HPVs. Secondly, the increasing risk for Ct values between 17 and 27 could be due to chance—the 95% confidence bands include a monotone decreasing curve for the full range of Ct values. Finally, it is possible that very high viral loads are indicative of a proliferative infection which is less likely to have resulted in CIN2+.

Whereas the increasing risk with increasing Ct values between 20 and 25–27 (Fig 4) is inconsistent with the underpinning logic that risk increases with viral load (which is inversely relative to Ct values), there are relatively few samples with (non-HPV16) Ct values in this range. Further, for both HPV18 and other HPVs, those with Ct <30 have greater probability of CIN2+ than those with Ct ≥30. For the endpoint CIN3+, this difference in risk is more pronounced for HPV18, but the risk is uniformly low for other HPVs. Further studies are required to see whether it would be safe to include all non-16/18 HPV samples in the low-risk category regardless of their Ct value. For now, we recommend caution and include non-16/18 HPV in intermediate risk if their Ct value is less than 30.

Given that HPV Ct values and HPV genotypes are directly available from many PCR-based (as well as isothermal amplification) HPV assays [15], our findings demonstrate the importance of HPV genotype and its Ct value for risk stratification of HPV–positive individuals. This is particularly valuable on self-samples for which reflex cytology is not available for triage. Integrating the HPV genotypes and Ct values into the clinical management process for cervical lesions has also been considered in cervical screening using clinician samples [11–15]. Direct referral to colposcopy of a small high-risk group could improve the efficiency of the clinical management process and minimize loss of follow-up in the referral pathway. For individuals at "intermediate risk," referral for a clinician-collected sample as in YouScreen seems appropriate. For the low-risk group, a repeat self-sample in 12 months would seem justified, but an alternative would be referral for a clinician-collected sample depending on the available healthcare resources and patient preference.

Self-sampling has been proven to be effective in enhancing the cervical screening coverage by reaching more of the under-screened population [7]. By enabling a more targeted management of HPV–positive individuals, healthcare systems can allocate resources more efficiently, potentially reducing the need for multiple healthcare visits and lowering the overall cost of cervical cancer prevention efforts. This approach could refine triage strategies, especially in low- and middle-income countries where access to quality cytology is limited [31]. It may also be an attractive option for individuals who are reluctant to have a clinician sample or who find a speculum examination painful.

The association between HPV Ct values, as a proxy for viral load, and the risk of cervical precancer and cancer has been investigated and proposed as a triage marker [11–15], but few studies have focused on self-samples. According to the cervical screening guidelines in United States, individuals who were HPV16/18–positive have been recommended to receive a direct referral to colposcopy [32]. The findings in our study are consistent with existing studies that HPV genotypes and HPV Ct values are associated with risk of high-grade cervical lesions [11–16]. A recent study from the Dutch national organised cervical screening programme investigating primary HPV testing on self-samples versus clinician-collected samples reported that high Ct values (>35) on self-samples were less likely to be associated with CIN2+ (and CIN3+) but low values (<30) did not have higher risk than intermediate (30–35) values [11]. It is possible that failure to distinguish HPV types in that study masked the full association with Ct values. Studies from New Mexico [12] and Costa Rica [16] concluded that HPV genotypes, in particular HPV 16, and viral load are significant predictors of CIN2+ and CIN3+. Several papers from the Chinese Multi-site Screening Trial have proposed HPV Ct value as a triage marker of high-grade cervical lesions [13–15]. A study based on self-samples showed that Ct values were associated with the severity of cervical lesions and suggesting an appropriate cut-off of 33.7 combined with HPV16/18 for triage of HPV–positive individuals [13]. In studies based on clinician-collected samples, studies reported that HPV16/18 genotyping combined with low Ct values for non-16/18 hrHPV, especially the A9 group (types 31, 33, 35, and 58), demonstrating satisfactory sensitivity and specificity for detecting CIN2+ or CIN3+ [15]. One

issue with cross-study comparison is that while general trends have emerged, different assays have been employed and Ct values between assays are not directly comparable. Ct values provide a proxy of load and the assays that include them may be considered semiquantitative rather than quantitative as there is no normalisation to adjust for cellular input. Advances in technology that provide absolute quantification—such as digital PCR, will provide further insight into the relationship between viral load and disease outcome, allowing for refinement in the development and application of HPV assays in a screening context.

With increasing impact of HPV vaccination in many countries, the risk of infection of vaccine-type HPVs and cervical lesions are expected to be reduced considerably [33,34]. In such a setting, referring individuals who test positive for HPV16/18 might be more efficient. However, the changes in HPV types causing pre-cancerous cervical lesions in the population might necessitate reassessment of clinical management and referral criteria for non-HPV16/18 types to account for the changing epidemiology. Cervical screening programmes will benefit from continuously evaluating the performance of HPV testing in vaccinated populations and optimising the triage pathway to ensure ongoing efficiency [35].

Our study's strengths lie in its large-scale evaluation of opportunistically offered self-sampling to non-attenders within an organised cervical screening programme. It means our findings have immediate clinical implications for pilot implementation programmes and screening studies. These results could be used to enhance screening efficiency, optimising triage protocols, and personalising clinical follow-up based on HPV genotype and Ct values. Limitations include that the colposcopy assessment was restricted to individuals who had abnormal cytology after positive results of both self-samples and clinician-collected samples. In this study, different HPV assays were used for primary screening and follow-up tests. Even in this large-scale implementation trial, we had limited numbers of CIN3+ detected limiting the power for subgroup analysis by HPV genotypes especially for HPV 18.

Future studies in large-scale cohorts incorporating long-term observation of self-samples are required to further examine and validate the long-term risk of CIN2+ and CIN3+ corresponding to different HPV genotypes and their Ct values. Validating the predictive value of Ct thresholds for different HPV types possibly in combination with other triage markers such as DNA methylation analysis would also be of great value [36,37]. As the Ct values are assay-specific, the threshold of Ct values for referral should be determined separately for each assay and preferably tested in various settings. Additionally, the integration of self-sampling together with triage based on HPV genotypes and their Ct values into existing healthcare systems needs to be carefully managed and monitored.

In conclusion, our findings highlight the potential value of integrating HPV partial-typing and Ct values to permit risk stratification of HPV–positive women. Such an approach may have immediate clinical application for self-sampling and could mitigate the loss to follow-up associated with triage strategies that necessitate a clinician-taken sample.

## Supporting information

**S1 Table. Baseline characteristics of study sample.**
(DOCX)

**S2 Table. Proportion of HPV positivity in clinician-collected samples, referral to colposcopy, CIN2+, and CIN3+, by risk groups and age.**
(DOCX)

**S1 Fig. Risk of CIN2+ and HPV Ct value.** The predicted probabilities were generated by running mean smoothing (running line), logistic regression model with natural cubic spline

(mkspline), natural cubic spline (spline), and fractional polynomial regression (fp); 95% CIs are illustrated for fractional polynomial regression (95% CI fp) and logistic regression model (95% CI mkspline). HPV, human papillomavirus; CI, confidence interval; CIN2+, cervical intraepithelial neoplasia grade 2 or worse.
(DOCX)

**S1 Dataset. Stata dataset containing 871 observations on 15 variables.** The dataset includes the Ct values for the 4 channels (HPV16, HPV18, HPV other, and beta-globin) as well as the result of the clinical HPV test and where available, the cytology and histology.
(DTA)

**S1 Codebook. Excel Worksheet listing the 15 variables in the Dataset together with a more detailed description and how they have been encoded (e.g., "1 = Yes; 0 = No").**
(XLSX)

**S1 Script. Stata "do-file" containing the code used to run the analyses in the manuscript using the data from "S1 Dataset."**
(DO)

**S1 YouScreen Trial Steering Committee. List of the members of the YouScreen Trial Steering Committee and their affiliations.**
(DOCX)

## Acknowledgments

We thank all the women and other people with a cervix who took part in the study and the staff at the participating general practices without whom this research would not have been possible.

## Author Contributions

**Conceptualization:** Peter Sasieni, Anita W. W. Lim.

**Data curation:** Hasit Patel, Alexandra Lawrence, Anita W. W. Lim.

**Formal analysis:** Jiayao Lei.

**Funding acquisition:** Peter Sasieni, Anita W. W. Lim.

**Investigation:** Peter Sasieni, Anita W. W. Lim.

**Methodology:** Jiayao Lei, Kate Cuschieri, Hasit Patel, Peter Sasieni.

**Project administration:** Katie Deats.

**Resources:** Katie Deats.

**Writing – original draft:** Jiayao Lei.

**Writing – review & editing:** Jiayao Lei, Kate Cuschieri, Hasit Patel, Alexandra Lawrence, Katie Deats, Peter Sasieni, Anita W. W. Lim.

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
