## [Editor Report · Decision Letter 0]

1 Aug 2024

Dear Dr Sasieni, 

Thank you for submitting your manuscript entitled "Human papillomavirus genotype and cycle threshold value from self-samples and risk of high-grade cervical lesions" for consideration by PLOS Medicine.

Your manuscript has now been evaluated by the PLOS Medicine editorial staff and I am writing to let you know that we would like to send your submission out for external peer review.

During our initial assessment, we were unable to locate any information in the Clinical Trial Registry entry regarding the investigation of Ct values. This has led us to wonder whether the analysis presented in the submission is a post-hoc analysis of the YouScreen trial data. If this is accurate, it would be essential to transparently report the analysis as such in the Abstract, Methods, and Title.

Please re-submit your manuscript within two working days, i.e. by Aug 05 2024.

Feel free to email me at atosun@plos.org or us at plosmedicine@plos.org if you have any queries relating to your submission.

Kind regards,

Alexandra Tosun, PhD

Associate Editor

PLOS Medicine

---

## [Decision Letter · Decision Letter 1]

28 Aug 2024

Dear Dr Sasieni,

Many thanks for submitting your manuscript "Human papillomavirus genotype and cycle threshold value from self-samples and risk of high-grade cervical lesions" (PMEDICINE-D-24-02470R1) to PLOS Medicine. The paper has been reviewed by subject experts and a statistician; their comments are included below and can also be accessed here: [LINK]

As you will see, the reviewers were positive about the manuscript. However, the statistical reviewer in particular commented that there are significant methodological issues that need to be addressed. After discussing the paper with the editorial team and an academic editor with relevant expertise, I'm pleased to invite you to revise the paper in response to the reviewers' comments. We plan to send the revised paper to some or all of the original reviewers, and we cannot provide any guarantees at this stage regarding publication.

We ask that you submit your revision by Sep 18 2024. However, if this deadline is not feasible, please contact me by email, and we can discuss a suitable alternative.

Don't hesitate to contact me directly with any questions (atosun@plos.org). 

Best regards, 

Alexandra 

Alexandra Tosun, PhD 

Associate Editor

PLOS Medicine

atosun@plos.org

Comments from the academic editor:

The premise is useful, and as a study that is heavily dependent on the analysis, the statistical issues raised are good ones that must be addressed for this piece to be of optimum value.

Comments from the reviewers: 

Reviewer #1: The authors describe: Human papillomavirus genotype and cycle threshold value from self-samples and risk of high-grade cervical lesions: a post-hoc analysis of a randomized controlled tria.

This is an interesting study in order to defin groups of screened women that may avoid the step via GP's in cervbical cancer screening with self sampling. The cohort is of sufficient size and the report is well written. 

However I have several commonts which i feel need to be addressed:

- The title is somewhat misleading as as it suggests that randomsation has anything to do with this study, while randomisation plays now part at all.

- The authors report a very large difference in HPV positivity between the self samples (SS) and the clinician collected (CCS) samples as about 50% of the SS positieve samples was negative on the CCS. This needs furhter explanation as it is of major influence on the results. Now the 50% saples that are negative on CCS lack any follow up data. so it remains unsure what cervical abnormalities are present in that group. This is a major drawback for the results and conclusions that may be drawn. Adding to this fact, also longer follow up data on women without direct referral after triage cytology are missing. 

- Additionally the manuscript lacks any explanation on this difference. Indeed other studies did find differences between HPV positivity in SS and CCS but not this large. 

- The manuscript also lacks description on the lab policy on the dry Floqswabs. In what median was the swab suspended. How many mls of it, as too little or too much dilution may directly influende the CT value of the PCR. In order to be able to repeat this experiment this information is crucial. I suspect rather low volume of suspension causing the much higher HPV positivity in the SS. 

- In the results both CIN 2+ and CIN3+ are reported. The major interest however should be in CIN 3+ as this is the real premalignancy. With that in mind number of women with CIN 3 in this study is relatively low so not many firm conclusions may result from this study. Especially the number of women > 50 years with CIN 3 is so low that no conclusion on age may be valid.

- The identification of a small group with HPV 16 and low CT values (5%) for direct referral is well described. Using the CT value as a surrogate viral load marker identifies a high risk group. However with direct referral one may argue that the colposcopist misses information that normally is present during colposcopy. Every colposcopist will prefer a cytology result, giving direction to what may be expected during colposcopy, and what management is expected, especially if abnormalities of the cylindric cells is suspected. So most colposcopist will still want a cytology report to assist them in colposcopic policy. 

- The main advantage of triage with CT value and genotyping is the identification of a subgroup that may be followed with a SS in 12 months time. But especially in this group viral load has not proven its value in the past. Many more studies like this need to be done in order to have sufficient information that CT values are valuable, as incidences of CIN 3+ may be low but need confirmation in larger cohorts. . 

- I miss any discussion on other markers than CT value or genotyping tha may be performed on SS. Methylation markers have been shown to be of value in this respect and combining genotyping, CT values and methylation markers may perform even better.

Reviewer #2: This research used data from the existing stepped-wedge YouScreen pragmatic trial to examine the association between test results from self-collected cervicovaginal samples and the results of a clinician-collected sample combined with any diagnosis of cervical intraepithelial neoplasia grade 2 or worse (where histopathological testing was completed). HPV type and cycle threshold (Ct) values ascertained from the self-collected sample were the specific biomarkers of interest. The results of these analyses were used to propose a risk stratification algorithm intended to refer similar proportions of individuals for further testing as the current, in-person standard of care. 

While the motivation behind and value of this research was well explained by the authors, there are substantial methodological issues that require addressing. Broadly, I don't yet understand why the authors chose the methodology employed i.e., developing a risk model that aims to refer in the same patterns as standard of care. The literature on clinical agreement and set-point ascertainment is well developed and it's unclear to me why it has not been leveraged here. Additionally, the statistical analysis methods presented don't yet have sufficient detail and rigour. No explanation was provided of why fractional polynomial regression (which is typically used for continuous endpoints) was used and how it was integrated with the use of logistic regression. Indeed, no details are provided about the modelling approach other than identifying the use of logistic regression. Given the unusual combination of statistical techniques, I would expect citations to reputable methodological prior work - the current citation to the statistical software manual is not sufficient. Finally, there are several apparent contradictions that are not explained, and some claims are made without sufficient support.

Detailed feedback on the manuscript is provided below. My primary focus has been methodological, but I have made a few broader points as well. All items are major unless indicated otherwise. 

1. General 

1.1 The manuscript title says the current study is based on a RCT, but the underlying trial is cluster-randomised/stepped-wedge. This may be misleading as "randomised clinical trial" is generally taken to mean individual, not cluster. 

2. Abstract

2.1 Please revise based on the provided feedback below. 

3. Introduction

3.1 The logic motivating the study (benefits of self-testing over clinician sample collection combined with prior research suggesting low Ct can be indicative of serious disease in some circumstances) was well presented but the research gap could be further argued. That is, has development of a risk stratification model in this context been attempted before? If so, how does this build on it? If not, say so. 

3.1 "Self-samples has also demonstrated either a comparable or slightly lower clinical sensitivity and specificity for detecting high-grade cervical lesions, providing a valuable tool for early detection and prevention of cervical cancer." The authors implicitly recognised the importance of sensitivity and specificity here, increasing my confusion as to why these and other clinical agreement statistics were not used in this research. 

3.2 HPV, cycle time, and self- vs. clinician-sampling are explained clearly and succinctly. 

3.3 The definition of study aims is clear, but I question below whether these aims are addressed with the current methodology and results. 

4. Material and methods

4.1 Study setting and population

4.1.1 [MINOR] I think it's worth noting that YouScreen was stepped-wedge.

4.1.2 [MINOR] Unclear why the boroughs of London are identified - this is not informative to global readers and doesn't appear to be pertinent to the current study. If I am missing something (such as these areas being particularly low or high socioeconomically) please clarify. 

4.1.3 In the final para the authors say "Ct values are not readily available from cobas 4800…" then "… we managed to extract Ct values…" and go on to describe in detail how the cobas 4800 produces Ct values and how they are central to how HPV subtypes are determined. This appears contradictory and should be clarified. 

4.2 Laboratory testing

4.2.1 This section is well-written and clear. I would only make the point that consistently in the text of the manuscript, the working hypothesis is that the lower the Ct value, the higher the viral activity (and, possibly, the higher the risk of serious disease). I will refer back to this point later. 

4.3 Statistical analyses

4.3.1 "These proportions were further stratified by age (categorized as <30, 30-49 and 50+), the overall HPV Ct value categories (<30, 30-35 and 35+)." Why/how were these cut-offs chosen? 

4.3.2 Please explain why fractional polynomial regression (which is typically used for continuous endpoints) is used here and how it is integrated with the used of logistic regression. I'm also curious why the authors didn't consider splines? 

4.3.3 Please detail the modelling approach thoroughly - how many models used and of what type? Covariates? Was goodness of fit assessed? If models are to be used to produce predicted probabilities, the modelling results must be presented too. 

4.3.4 Given the unusual combination of statistical techniques, I would expect citation(s) to reputable methodological prior work - the current citation to the statistical software manual is not sufficient. 

4.4 Target risk classification

4.4.1 The methodological basis of this approach to defining risk strata is not presented. The authors must establish the validity of the presented approach where Ct cutoffs within HPV types are tweaked until the expected referral patterns broadly match the referral patterns observed in standard of care. Again, there are established approaches to cutoff ascertainment and judging the success (or otherwise) of risk algorithms that could be leveraged. 

4.4.2 "In the HPV primary screening pilot, among those referred "immediately" with HPV positive and abnormal cytology, about 40% (3,060/7,542) had CIN2+." Later in the manuscript you refer to a PPV cut-off of 40%. If this is the 40% you are referring to please make that clearer here. Put another way, you don't say anywhere in the methods you are calculating PPVs but rely on them later. 

6. Results

6.1 Study Population

6.1.1 [MINOR] "hrHPV" = "high risk HPV"? 

6.1.2 1,001 of 7,643 patients in text but 1,002 of 8,340 patients in figure. 

6.2 HPV and beta-globin Ct values

6.2.1 "Notably, there were many outliers (Ct >32.5) for beta-globin Ct values, indicating samples with low levels of human DNA, in all groups." This is identified but not explored further. Is this expected? Does this cast doubt on the reliability of the samples and subsequent findings? Or is it not problematic because it appears evenly distributed? Consider clarifying here or in Discussion. 

6.3 HPV types, HPV Ct values and risk of high-grade cervical lesions

6.3.1 Throughout this paragraph the authors refer to "risk" interchangeably with proportions of patients. I understand why this approach was taken but I would recommend against it because it has the potential to mislead - these are simple proportions of a small number of patients, not modelled risks with reported uncertainty. 

6.3.2 "Comparing by HPV Ct values, we observed a clear gradient of decreased risk of HPV positivity on clinician-collected samples with increasing Ct values…" There is currently insufficient evidence presented to make this claim - the patient numbers are small, no analyses or statistics are presented (just simple mean percentages), and the authors are relying on three levels only. 

6.3.3 "The risk of CIN2+ plotted as a smooth function of the HPV Ct value is shown in Figure 3a. The risk of CIN2+ is below 10% for Ct values below 20." Notwithstanding the methodological concerns already raised in 4.3, this is not justified by the existing analyses. The number of patients is very small and the presented confidence intervals take in the entire range of risk values. 

6.3.3 "The risk of CIN2+ is below 10% for Ct values below 20. It increases to nearly 15% for Ct-values in the mid-20s and then gradually falls dipping below 5% at about 35 and below 2.5% by 40." This result is inconsistent with a key hypothesis underpinning this research, that low Ct corresponds to higher viral load and, consequently, higher risk of severe disease. It is crucial this inconsistency is addressed. Any explanation must also incorporate any implications of this statement from the Laboratory Testing section of the Introduction "The manufacturers have defined Ct value-cutoffs for all the HPV channels to determine positive results (the Ct value must be below the cut-off for the test to be called positive), which are 40.5 for HPV16 and 40 for HPV18 and the pooled 12-HPV channels.". 

6.4 Risk score

6.4.1 Please refer to feedback 4.4 above - I would reiterate the importance of clearly articulating in the Methods analyses that produce statistics presented here (i.e., PPV). 

6.4.2 This section would be strengthened through the quantification of uncertainty (confidence intervals at a minimum). 

6.4.3 The use of "Low Ct is riskier" logic underpinning the intermediate and low risk strata for HPV non-16/18 (i.e., outside of HPV-16) appears inconsistent with the bell-shaped risk curves presented in Figure 3a and 3b (right panel). 

6.4.4. "Using this classifier allows those at greatest risk to be sent directly to colposcopy and detects three quarters of CIN2+ immediately whilst allowing half of individuals to be managed by repeat self-sampling." There is not yet sufficient evidence presented to justify this statement. 

7. Discussion

7.1 I will not repeat feedback already provided when similar feedback applies to the discussion, but please ensure any changes made in the earlier sections are reflected here too. 

7.2 "However, we think that dividing HPV positive individuals into four risk groups may risk overfitting these data and we prefer to divide into three risk groups as described in the results section." I would suggest against the use of the word "overfitting" here. It has specific meaning with respect to statistical modelling, and there is no model being fit here - simply a partitioning of patients into risk strata. 

7.3 "It means our findings have immediate clinical implications…" Such a strong statement is not yet justified by the results; indeed, the authors go on to say "Future studies in large-scale cohorts incorporating long-term observation of self-samples are required to further examine and validate the long-term risk of CIN2+ and CIN3+ corresponding to different HPV genotypes and their Ct values."

8. Tables and Figures

8.1 I will not repeat feedback already provided when similar feedback applies to the tables and figures, but please ensure any changes made in the earlier sections are reflected here too.

8.2 I suggest adding a table characterising the sample of participants. 

8.3 Figure 1 slightly differs from similarly worded parts of the original YouScreen trial's CONSORT diagram. I suggest making clear why there are differences. 

8.4 Also in Figure 1, I am not sure what the footnote refers to or is trying to communicate. Suggest clarifying. 

Reviewer #3: This is post-hoc analysis of a large cervical cancer screening trail. The trail investigated the utility of HPV testing in combination with self-sampling. The aim of this analysis was to determine if CT values in combination with HPV genotype, as detected on self-sampling, can help to predict the ultimate diagnosis of CIN2+lesions when using physician collected samples and routine referral for cytology and colposcopy as reference standard. The ultimate aim is to reduce LTF of high-risk participants by identifying a high risk screen-positive group that could be referred directly to treatment without cytology triage.

The methodology is well motivated and explained. It is s a pity that the 172 women with positive HPV and normal cytology could not be assessed after their recall, however, that does not take away from the clear demonstration of CT value and genotype to guide clinical action. The three management groups, as described, had significantly different risks for the detection of CIN2+. 

It seems that the histology results were taken from routine clinical datasets. Perhaps the authors can add a sentence to explain that histology was reported in a "real-world" situation and that there was no central pathology review. 

Figure 3 is a good visualization of how CT value can add diagnostic accuracy, particularly in those that test positive for HPV type 16.

This paper adds value to our understanding of how CT values and HPV genotype hierarchy can help to stratify risk, even in self collected samples.

---

* Please upload any figures associated with your paper as individual TIF or EPS files with 300dpi resolution at resubmission; please read our figure guidelines for more information on our requirements: http://journals.plos.org/plosmedicine/s/figures. While revising your submission, please upload your figure files to the PACE digital diagnostic tool, https://pacev2.apexcovantage.com/. PACE helps ensure that figures meet PLOS requirements. To use PACE, you must first register as a user. Then, login and navigate to the UPLOAD tab, where you will find detailed instructions on how to use the tool. If you encounter any issues or have any questions when using PACE, please email us at PLOSMedicine@plos.org.

FIGURES AND TABLES

SUPPLEMENTARY MATERIAL

REFERENCES

* Where website addresses are cited, please include the complete URL and specify the date of access (e.g. [accessed: 12/06/2024]).

STUDY TYPE-SPECIFIC REQUESTS

* Please be explicit in the title and the abstract that is a post-hoc sub-study/follow-on study of the primary trial. We suggest reporting in-line with CONSORT explicitly stating the sub-study nature and ensuring that the abstract details the main trial items in 2-3 sentences, including the study population, dates, intervention and primary outcome. The majority of the abstract should then describe the complete details of this post-hoc sub-study. 

* Please complete the CONSORT checklist and ensure that all components of CONSORT are present in the manuscript as well as clearly defined details of this sub-study. When completing the checklist, please use section and paragraph numbers, rather than page numbers as these often change in the event of publication.

* Please ensure that study registration details are included in the Methods section.

* Abstract: Please include the study design, population and setting, number of participants, years during which the study took place (original enrollment and follow up), length of follow up, and main outcome measures.

* Please include absolute numbers wherever you report percentages; eg, n/N (%)

* In keeping with our commitment to Open Science, please include the study protocol document and analysis plan (including any amendments) as Supporting Information to be published with the manuscript if accepted.

---

## [Decision Letter · Decision Letter 2]

30 Sep 2024

Dear Dr. Sasieni,

Thank you very much for re-submitting your manuscript "Human papillomavirus genotype and cycle threshold value from self-samples and risk of high-grade cervical lesions: a post-hoc analysis of a modified stepped-wedge implementation feasibility trial" (PMEDICINE-D-24-02470R2) for review by PLOS Medicine.

Thank you for your detailed response to the editors' and reviewers' comments. I have discussed the paper with my colleagues and the academic editor, and it has also been seen again by two of the original reviewers. The changes made to the paper were mostly satisfactory to the reviewers. As such, we intend to accept the paper for publication, pending your attention to the reviewers' and editors' comments below in a further revision. When submitting your revised paper, please once again include a detailed point-by-point response to the editorial comments.

[LINK]

In revising the manuscript for further consideration here, please ensure you address the specific points made by each reviewer and the editors. In your rebuttal letter you should indicate your response to the reviewers' and editors' comments and the changes you have made in the manuscript. Please submit a clean version of the paper as the main article file. A version with changes marked must also be uploaded as a marked up manuscript file. Please also check the guidelines for revised papers at http://journals.plos.org/plosmedicine/s/revising-your-manuscript for any that apply to your paper.

We ask that you submit your revision within 1 week (Oct 07 2024). However, if this deadline is not feasible, please contact me by email, and we can discuss a suitable alternative.

Please do not hesitate to contact me directly with any questions (atosun@plos.org). If you reply directly to this message, please be sure to 'Reply All' so your message comes directly to my inbox.

We look forward to receiving the revised manuscript.

Sincerely,

Alexandra Tosun, PhD

Associate Editor 

PLOS Medicine

plosmedicine.org

Requests from Editors:

GENERAL COMMENTS

1) When presenting age, please add a unit, such as ‘years’. Please revise throughout the manuscript, including tables and figures (including those in the Supporting Information).

2) Please check whether the details provided on lines 658-686 match the information provided in the online submission form.

3) Thank you for your response regarding the CONSORT checklist. We will not be requiring you to complete the checklist.

4) There is no need to include the main trial protocol as a Supporting Information file, and since there was no formal study protocol document for the post hoc analysis, you do not need to include any document.

DATA AVAILABILTY

Please note that the data availability statement as written on lines 659-663 does not meet our journal requirements and does not match the statement provided in the online submission form. I have noted your comments in the point-by-point response and appreciate your efforts to provide the underlying data. Before resubmitting, please ensure that the paper complies with the PLOS Data Availability Policy.

For each data source used in your study: 

CODE AVAILABILITY

Please add a sentence to your data availability statement regarding the code used in the study, e.g. "The code used in the analysis is available in the Supporting Information files.” and add the file accordingly.

ABSTRACT

ll.52ff: Please report statistical information as follows to improve clarity for the reader "22% (95% CI [13%,28%]; p</=). For example, line 52-53: “44/855 (5%) were high-risk of whom 43% (19/44, 95% confidence interval [29.7%, 57.8%]) had CIN2+.” When reporting 95% CIs please separate upper and lower bounds with commas instead of hyphens as the latter can be confused with reporting of negative values.

AUTHOR SUMMARY

1) l.70: Please change to “endocervical samples”.

2) l.72: Please temper claims of primacy of results by stating, "to our knowledge" or something similar or remove the word “novel”.

INTRODUCTION

l.114: Please define ‘UK’ at first use.

METHODS AND RESULTS

1) l.198: Please define ‘hrHPV’ at first use. Once you have introduced the abbreviation, please use the abbreviation throughout the manuscript. We have noticed that you alternate between using the abbreviation and "high-risk HPV".

2) ll.311-314: Please revise the sentence. Do the three percentages refer to Ct values below 30, 30-35 and above 35?

3) l.324: Please change to “showed”.

4) “A further 11% (=93.2%-81.8%)” – We are not sure that the calculation is very clear. We suggest writing "(=5 individuals; 41/44, 93.2%)".

5) l.346: Please change to “were”.

6) ll.349-351: Please revise for clarity.

7) Figure 1: Please define ‘hrHPV’, ‘CIN2’, ‘CIN3+’. Please note that the figure itself does not include the footnote 'a)'.

8) Figure 2: Please remove the definition "CIN2+, cervical intraepithelial neoplasia grade 2 or worse; CIN3+, cervical intraepithelial neoplasia grade 3 or worse" below the figure as you seem to be differentiating only between normal and abnormal cytology. Please explain in the figure description what 'Cyt+' and 'Cyt-' refer to. We also suggest adding an explanation that beta-globin serves as an internal control for sample cellularity and which HPV types are included. Please remember that all figures and/or tables should be self-explanatory on a stand-alone basis.

9) Table 1: Please add a unit for age. Please note that footnote 'b)' is marked 'y' in the table.

10) Table 2: Please note that there are no footnotes in the table. Please add the risk group definitions below the table.

11) Figure 3: Please change to “All samples” and “Risk of CIN2+ and HPV Ct value in all samples”.

DISCUSSION

ll.371-372: Please revise for clarity.

REFERENCES

1) Please use the word “accessed” instead of “cited” when specifying the date of (e.g. [accessed: 10/04/2024]).

2) Please ensure that journal name abbreviations match those found in the National Center for Biotechnology Information (NCBI) databases (http://www.ncbi.nlm.nih.gov/nlmcatalog/journals), and are appropriately formatted and capitalized. For example, “New England Journal of Medicine” in reference [5] should be “N Engl J Med”.

SUPPLEMENTARY MATERIAL

In the published article, supporting information files are accessed only through a hyperlink attached to the captions. For this reason, you must list captions at the end of your manuscript file. You may include a caption within the supporting information file itself, as long as that caption is also provided in the manuscript file. Do not submit a separate caption file.

When SI files are contained with a single file:

Please label the file as ‘S1 Supporting Information’.

Please apply alphabetical labelling to each table and figure contained within the S1 file. For example, ‘Fig A’ to ‘Fig Z’ and ‘Table A’ to ‘Table Z’.

Plain text does not need to be labelled and can just be given a title as necessary. For example, ‘Statistical Analysis Plan’.

Please cite tables/figures as ‘Fig A in S1 Supporting Information’ and/or ‘Table A in S1 Supporting Information’, for example.

Please cite plain text as, ‘Statistical Analysis Plan in S1 Supporting Information’, for example.

When SI files are uploaded as separate files:

Please label tables as ‘S1 Table’ (so on) and figures as ‘S1 Fig’ (and so on).

Any additional documents (protocols/analysis plans etc.) can be labelled as ‘S1 Protocol’, for example. Please cite items as exactly as labelled.

SOCIAL MEDIA

To help us extend the reach of your research, please provide any X (formerly known as Twitter) handle(s) that would be appropriate to tag, including your own, your co-authors’, your institution, funder, or lab. Please enter in the submission form any handles you wish to be included when we post about this paper.

Comments from Reviewers:

Reviewer #1: Comments have well been answered and adjusted

Reviewer #2: Thanks to the authors for the clear and cogent responses to my feedback and associated changes to the manuscript. In most cases I agree and consider the items closed (assume this is the case unless otherwise indicated), but I have identified some follow-up points for further consideration that I have detailed below. In several cases the response to my question is clear but has not yet been reflected in the manuscript. In a small number of cases I have provided further suggestions. 

Before providing the detailed follow-up points, I did want to make one general observation which is easy to fix and, on reflection, caused me quite a bit of confusion. When I read "risk" or "risk score", I immediately think of a continuous risk percentage between zero and one. Conversely, "risk category" or "risk strata" refer to ordered categories of risk. Given you are presenting both risk percentages (CIN2+) and risk strata (High/Medium/Low), I suggest choosing your preferred naming and ensuring it is used consistently throughout the manuscript text and tables/figures. This is of particular importance for more methodologically inclined readers because, as you know, choice of statistical model and approach hinges on the outcome variable type. 

ADDITIONAL FOLLOW-UP POINTS

"We are following Castle and Katki (Nat Rev Clin Oncol 2016) who advocate 'equal management of equal risks' as a framework for developing screening guidelines and practice. It is precisely our different perspective that distinguishes this paper from the existing literature which focuses on sensitivity and specificity."

These sentences in the authors' response address my concern about choice and justification of methodology. I suggest this is reflected in the Introduction, including citing the work by Castle and Katki identified by the authors. 

Point 4.3.3.[a] The authors' response is very clear and helpful and, I believe, essential to any reader wishing to understand or replicate the analyses. As such, I would suggest this level of detail is included in the manuscript or (if preferred/required due to word count), in the supplementary materials. 

Point 4.3.3.[b] Further to the author's point "Note that for the primary model, there is no need for logistic regression, we can simply look at the observed proportions with CIN2+ in each group and calculated 95% confidence intervals using Wilson's method." I understand the primary analyses (as reflected in Table 1) and see how they are used in the definition of the three risk strata. That said, I am still unsure why the authors used this complicated combination of strata and descriptive univariate calculations when four multivariable models (one for each of the four binary endpoints represented by columns in Table 1) or, for that matter, a single model with an ordinal outcome, would have been simpler and allowed for assessment of relative salience of the various factors i.e., age group, HPV type, Ct value, and interactions thereof. That said, given the small number of patients and purpose of these analyses (informing the development of the risk strata as opposed to definitively establishing risk percentages), I accept the current analyses are sufficient. 

Point 4.4.2. I am not convinced you have addressed my point here. In the results you still refer to "PPV" (lines 348 and 357) and it is not included in your methods; either as a calculation using these data or as a reference point sourced from prior work. Relatedly, I really am struggling to understand what you are trying to communicate here given your response to point 4.4.2 suggests you believe PPV is not useful in and of itself as a comparator, but you continue to use it in this way in the manuscript. 

6.1.1 Perhaps I am missing it, but "hrHPV" does not yet appear to be defined in the manuscript text. 

6.2.1 "Although values over 32.5 are outliers on the box and whiskers plots, they were all within the manufacturer's adequate range." Suggest including this (or similar) in your discussion given you present the results. 

6.3.1 The changes read very well. 

6.3.2 This is OK, but I still have a concern about the use of the phrase "clear gradient" given gradient means gradual change and you only have three levels. 

6.3.3 & 6.4.3 These responses are excellent. Well done. 

6.4.4 & 7.3 Constraining your results to informing future pilot implementation programs and screening studies (as opposed to definitively establishing risk of outcomes or immediately changing current clinical practice) addresses my concerns here and elsewhere. 

Figure 2 notes include definitions for CIN2+ and CIN3+ but these terms are not included in the title or figure itself.

[LINK]

General Editorial Requests

---

## [Decision Letter · Decision Letter 3]

21 Oct 2024

Dear Dr Sasieni, 

On behalf of my colleagues and the Academic Editor, Elvin Hsing Geng, I am pleased to inform you that we have agreed to publish your manuscript "Human papillomavirus genotype and cycle threshold value from self-samples and risk of high-grade cervical lesions: a post-hoc analysis of a modified stepped-wedge implementation feasibility trial" (PMEDICINE-D-24-02470R3) in PLOS Medicine.

I appreciate your thorough responses to the reviewers' and editors' comments throughout the editorial process. We look forward to publishing your manuscript, and editorially there are only a few remaining minor stylistic/presentation points that should be addressed prior to publication. We will carefully check whether the changes have been made. If you have any questions or concerns regarding these final requests, please feel free to contact me at atosun@plos.org.

Please see below the minor points that we request you respond to:

1) Abstract: In the last sentence of the Abstract Methods and Findings section, please describe the main limitation(s) of the study's methodology.

2) Table 1: Please note that the word colposcopy in the second column heading is missing a "y". Please revise.

3) Table 2: Please note that in the table itself, footnote b is included twice and footnote c is missing. Please revise.

4) Please include the information provided in the Data Sharing Agreement on lines 676-679 in the Data Availability Statement in the online submission form.

Before your manuscript can be formally accepted you will need to complete some formatting changes, which you will receive in a follow up email (including the editorial points above). Please be aware that it may take several days for you to receive this email; during this time no action is required by you. Once you have received these formatting requests, please note that your manuscript will not be scheduled for publication until you have made the required changes.

PRESS

Sincerely, 

Alexandra Tosun, PhD 

Associate Editor 

PLOS Medicine